bioengineering/biomechanics/computational biology

cell mechanobiology, cell bulk elasticity, migratory index, cell migration, cell invasion, cancer cells

**Author for correspondence:**
Muthukumaran Packirisamy
e-mail: pmuthu@alcor.concordia.ca

# Cancer cells optimize elasticity for efficient migration

Ahmad Sohrabi Kashani and
Muthukumaran Packirisamy

Optical Bio-Microsystem Lab, Micro-Nano-Bio-Integration Center, Department of Mechanical, Industrial and Aerospace Engineering, Concordia University, 1455 De Maisonneuve Boulevard West, Montreal, Quebec, Canada H3G 1M8

 ASK, 0000-0002-3673-4132; MP, 0000-0002-1769-6986

Cancer progression is associated with alternations in the cytoskeletal architecture of cells and, consequently, their mechanical properties such as stiffness. Changing the mechanics of cells enables cancer cells to migrate and invade to distant organ sites. This process, metastasis, is the main reason for cancer-related mortality. Cell migration is an essential step towards increasing the invasive potential of cells. Although many studies have shown that the migratory speed and the invasion of cells can be inversely correlated to the stiffness of cells, some other investigations indicate opposing results. In the current work, based on the strain energy stored in cells due to the contractile forces, we defined an energy-dependent term, migratory index, to approximate how changes in the mechanical properties of cells influence cell migration required for cancer progression. Cell migration involves both cell deformation and force transmission within cells. The effects of these two parameters can be represented equally by the migratory index. Our mechanical modelling and computational study show that cells depending on their shape, size and other physical parameters have a maximum migratory index taking place at a specific range of cell bulk elasticity, indicating the most favourable conditions for invasive mobility. This approximate model could be used to explain why the stiffness of cells varies during cancer progression. We believe that the stiffness of cancer or malignant cells depending on the stiffness of their normal or non-malignant counterparts is either decreased or increased to reach the critical condition in which the mobility potential of cells is approximated to be maximum.

## 1. Introduction

Over two past decades, many researchers have revealed that mechanical properties at the cellular level, mechanobiological

properties, can provide diagnostic information on various abnormalities like cancer. Cells are the smallest, dynamic and functional living unit of human organs and many diseases like cancer cells originate from alternations at the cellular level [1,2]. There is mounting evidence suggesting that biological functions of cells can be correlated to the mechanical properties [3–5], and this growing knowledge has motivated many bioengineers to investigate how physical and mechanical properties of cells contribute to migration, progression and invasiveness of cancers. Alternation in mechanobiological properties of cells will influence functional characteristics of cells such as cellular proliferation, progressions, growths, mobility, invasiveness and differentiation [6]. Living cells possess different mechanobiological properties such as stiffness and viscosity, which have been found to be significant mechanical biomarkers for cancer detection at early stage. Stiffness is a fundamental mechanical property of structures or components and is defined by material properties such as Young's modulus and geometrical configuration [7,8]. Several scientific surveys have shown that cancerous and invasive cells are softer compared with non-invasive and healthy cells [9]. However, paradoxically, their tumours, including the surrounding environments of cells, are more rigid compared with cells as cancer cells undergo mesenchymal–epithelial transition [10]. Deformability coupled with less adhesion in cancer cells enables them to change their positions within body tissues through cell migration and invasion [11,12]. This phenomenon, metastasis, is the main threat of cancer to human health. According to the most reported results on the mechanics of various cancers, softness is a striking characteristic of cancer cells, and its degrees could give an estimation of cancer migration and invasiveness [3]. For example, it has been reported that the metastatic activities of human breast cancer cells can be increased by lowering the stiffness of cells [13,14]. Findings of Prabhune *et al.* [15] showed that the actin organization of malignant thyroid cells is disrupted, giving the malignant cells two to five times smaller bulk Young's modulus in comparison with the primary normal thyroid cells. In other work, Xu *et al.* [9] used atomic force microscopy (AFM) technique to examine the stiffness of highly invasive ovarian cancer cells (HEY A8) in comparison with benign ovarian epithelial cells (HEY), suggesting that malignant cells display a lower intrinsic stiffness than non-malignant cells by remodelling their actin organizations. These findings suggest that invasiveness and migration speed of cells can be related to the softness level of cells, and this correlation it is believed can be used to estimate the invasiveness and migration speed of cancers [16]. Despite many findings on the softening of cancer cells and their subsequent increase in the migratory and invasive capability, it is not appropriate to draw a firm conclusion about the invasiveness and the softening of different types of cancers [13,17]. There are a few contradictory results in which it was shown that cancerous and invasive cells are stiffer compared with normal and non-invasive cells [17]. Palmieri *et al.* [18] showed exactly a reverse stiffness–invasiveness relationship. Their AFM measurements confirmed that, in colorectal cancer cells, non-invasive (R-type) cells have smaller Young's modulus compared with the invasive cells (E-type). In another work performed by Bastatas *et al.* [19], it was also revealed that the stiffness of invasive or metastatic prostate cancer cells is significantly increased compared with their normal counterpart cells. Based on these results, invasive or cancerous cells are not necessarily softer compared with their non-invasive and healthy counterpart cells. It is thought that invasive cancer cells could be either stiff or soft. Stiff cancer cells cannot deform themselves and soft cancer cells cannot generate and transmit forces to squeeze through pores of highly cross-linked collagen fibres in their microenvironment [20]. Figure 1 schematically shows how relatively stiff or soft cancer cells can move into their local microenvironment. For a relatively stiffer environment, the cancer cells need to be deformed largely to squeeze through the microenvironment, while cancer cells need to be relatively stiffer to move into the softer microenvironment.

Similar contradictory results might also be observed in different responses of cancer cells to anti-cancer drugs. Many anti-cancer drugs, like chemotherapeutic agents, are purposely designed to target the cytoskeleton and mechanics of cancer to control cell migration and consequently cell metastasis [6,21]. Some studies have shown that treating cells with anti-cancer drugs or nanoparticles regulated the progress and migration of cancer by reducing their stiffness values [22,23], while other investigations are suggesting exactly an inverse relationship, treated cells are stiffer and less invasive and motile compared with the non-treated cells [24,25].

Looking at these inconsistent results, some fundamental questions might be raised on the reliability of cell stiffness measurements to evaluate the invasiveness and migration of cells and the efficacy of anti-cancer drugs designed to target the cellular structures. How could cells' stiffness be used to predict the migratory and invasive ability of cells? If cancerous and invasive cells are softer compared with non-invasive and healthy cells, is it always effective to design anti-cancer drugs to increase the cell stiffness to control the migration and invasiveness of cancer? It seems that cell stiffness alone is not capable of answering these key questions. In the process of metastasis, cell migration is essential to regulate many biological processes required for cancer invasion, and invasive cells are more migratory than non-invasive cells

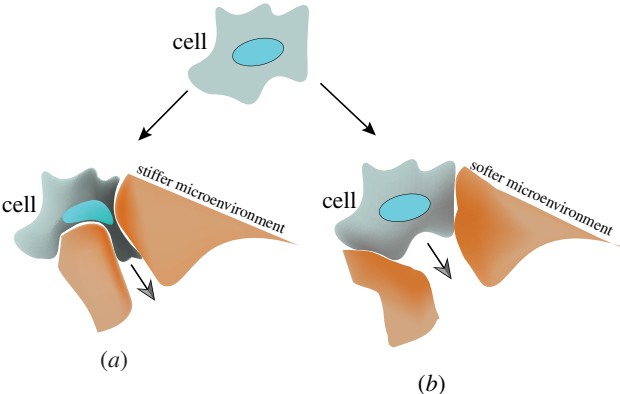

**Figure 1.** Cancerous cells can be relatively either softer or stiffer compared with their microenvironment. Depending on the rigidity of cells and their microenvironment, (*a*) they can be deformed or (*b*) they can deform their microenvironment.

[26–30]. For example, as mentioned earlier, under some treatments, the invasive ability of invasive cancers is reduced along with a reduction in their migratory ability [23,24]. In the context of cell mechanics, force transmission and deformation of cells are influenced by the elasticity of their bodies and are essential for cell migration [31–33]. To study and predict the migratory behaviour of cells under different microenvironments, we need to model both force-based migration and deformation-based migration through cancer mechanobiological behaviour in a unique way. Once a force is generated in cells, it is propagated throughout the cell body. Depending on the structural and material properties of cells, the body of cells is deformed, and a portion of that force reaches the cell–microenvironment contact points, trying to overcome resisting forces at those points and making cells move forward. All these phenomena, force generation and transmission, and the subsequent deformation are energy-dependent processes. Once cells are deformed due to the internal force, mechanical energy is stored in cells. The stored energy can be changed to other forms of energy, and under ideal conditions could approximate the potential of the system (cell body) to do work (cell movement). Under ideal conditions, the stored energy could be used to represent both deformation and force transmission in cells. Although cell migration and cell invasion are complex phenomena, with the energy model, the migration behaviour of cancer cells could be approximated to compare their migratory abilities globally based on their stiffness values. It is thought that the energy model can answer mechanobiological questions about stiffness–migration interactions.

In the current work, we developed a computational study to test our hypotheses about the stored energy potential in cancer cells due to the internal force and approximate the migratory potential of cancer cells. The higher energy potential implies increased migration as this energy of cells is transformed either through deformation or force and could represent the degree of migration and be called 'migratory energy'. Our objective is to use computational techniques to approximate the cells' migratory ability during epithelial–mesenchymal transition [10] in the absence of biological contributions from cell and microenvironment that are negligible for this study. We modelled a single cell and studied its migratory energy by varying its bulk stiffness to show how cell mechanics influence the cells' migratory ability. Based on this approximate model, depending on the shape and size of cells, there is a critical range of stiffness in which their migration is predicted to be high. Beyond that range, cells might lose their capability to migrate fast and have less migratory potential to move into other parts of the body. It is believed that healthy and non-invasive cells possess a bulk stiffness distant from the critical range, while cancerous and invasive cells tend to modulate their stiffness to a value in the critical range to maximize their migratory potential required for cancer progression. Our hypothesis and the approximate model could accurately interpret the earlier-mentioned contradictory results and shed light on the reported results about cell stiffness and their migratory and invasive potentials.

# 2. Material and methods

## 2.1. Rigidity index

Some cancer and invasive cells could be softer than their normal and non-invasive counterparts, while a few other cancerous and invasive cells are stiffer than their healthy and non-invasive

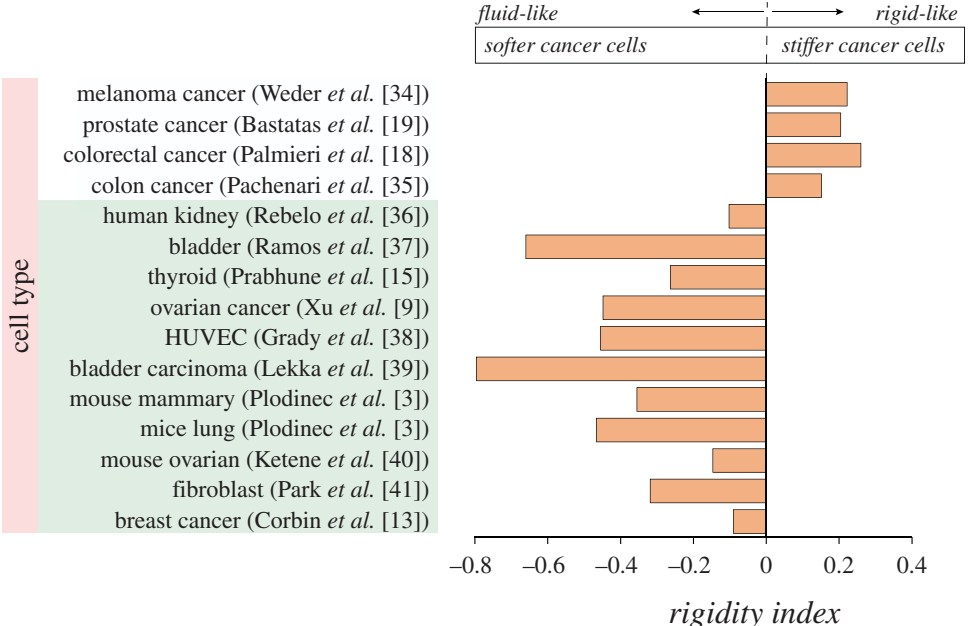

**Figure 2.** Increasing or decreasing the stiffness of cancer cells could enhance the invasive and migratory abilities of cells. Those changes might have happened either during cancer progression or once they are treated with different drugs. Cells stiffness values have been measured with different methods, stiffer: AFM [18,19,34], micropipette aspiration [35], softer: AFM [3,9,13,15,36–41].

counterparts. Similar behaviour also has been observed in the treated cells, showing that the cell migration speed and invasiveness can be reduced either by increasing or decreasing the rigidity of cells. These different behaviours could be presented with the rigidity index. For cancer or malignant cells, the rigidity index could be defined with respect to the elasticity of their normal or benign counterparts as follows:

$$r = \log_{10}\left(\frac{E_c}{E_n}\right),\tag{2.1}$$

where $r$ is the rigidity index, $E_c$ is the elasticity of cancer or malignant cells and $E_n$ is the elasticity of normal or benign counterparts with the same unit. The variation of $r$ is given in figure 2 for different types of cancer. This figure shows the rigidity index for different cancer types reported in the literature. For fluid-like cancer cells that are softer than normal cells, the rigidity index is negative ($E_c < E_n$), and for rigid-like cancer cells that are stiffer than normal cells, the rigidity index is positive ($E_c > E_n$).

## 2.2. Cell modelling and force generation

Cells have a heterogeneous complex material, and each region of cells has different mechanical properties and stiffness. Although the bulk stiffness of cells might change during the migration, the initial bulk stiffness was assumed to be constant to simplify the modelling and approximate the migratory potential of cells. Furthermore, to approximate the migratory ability of cells, we assumed that the contractile force and the bulk stiffness of cells are independent. Considering these assumptions, cells were modelled as a pure mechanical body with a bulk stiffness [31]. In this study, as shown in figure 3, the body of cancer cells was discretized into two zones; the nucleus with bulk stiffness of $K_1$ and the cytoplasm with bulk stiffness of $K_2$. The nucleus of cells is 4–10 times stiffer compared with the cytoplasm [42–44]. The shape and dimensions of cells in this study were assumed to be similar to human breast cancer cells described in [42]. The microenvironment of cancer cells is modelled with a substrate (with specific elasticity), and cancer cells are linked to their microenvironment (substrate) with a number of connections (adhesion complexes) as shown in figures 3c and 4a. For migration, cells need to generate a force to deform cells and transmit that force to the cell–substrate interface to overcome moving-resistance forces and make cells move forward. These two actions are dependent on the rigidity of cells in different regions. The internal force for the movement of cells is generated

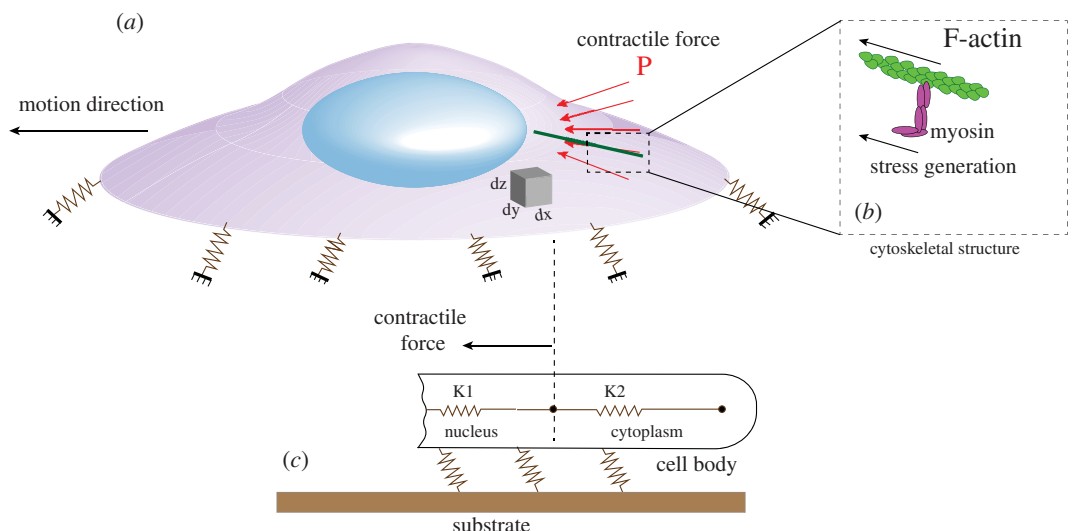

**Figure 3.** A schematic of a single cell showing how contractile force (*a*) is applied for migration in cells due to stresses generated in myosin network (*b*). The cell is attached to the substrate (*c*), and different regions of cells have different elasticities which can be simulated with a spring with a specific stiffness.

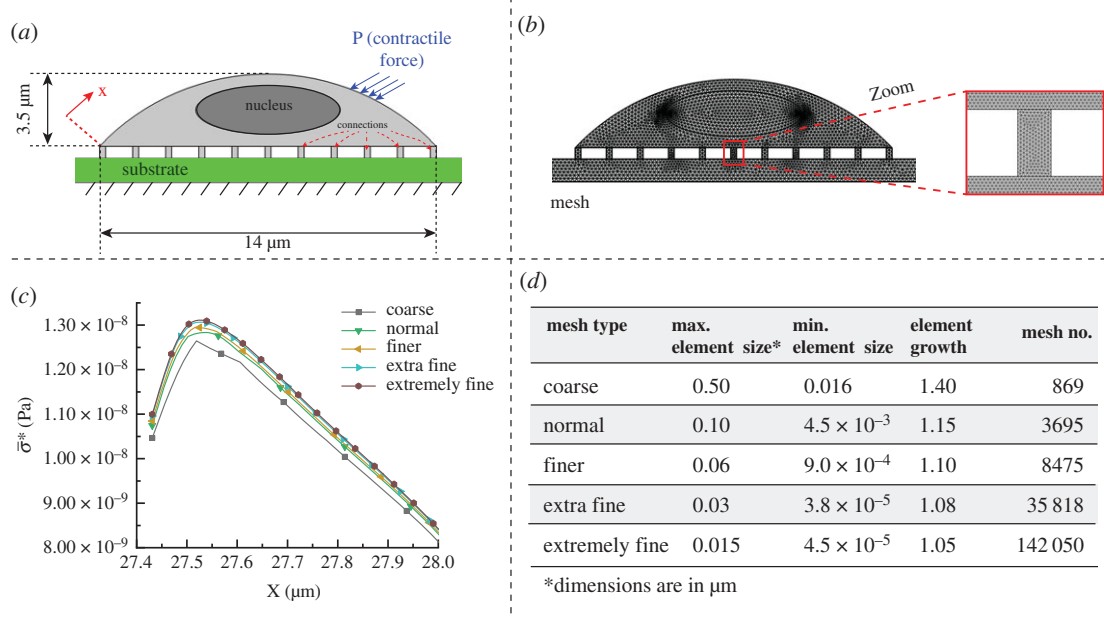

**Figure 4.** (*a*) Cell modelling in COMSOL, showing where forces are applied, (*b*) mesh elements distributions for the computational study, (*c*) mesh dependency study by measuring $\sigma^*$ on the top of cells for different mesh types and (*d*) different parameters used for different mesh types.

by polymerization of actin-filament and the molecular interactions between F-actin and non-muscle myosin II at the rear-middle of the cells [45] (figure 3*b*). The generation of the internal force occurs by converting chemical energy to mechanical work. In our simulation, a force at the rear-middle of cells was considered as the contractile force to quantify cell deformation and stress transmission within cells bodies (figure 3*a*).

## 2.3. Strain energy and migratory energy

As already mentioned, the two parameters, namely deformation and transmitted force, play pivotal roles in the movement of cells [45,46]. Displacement field can be calculated in different regions of cells by

considering the deformation throughout the cells, and the propagated force in each region can be quantified by computing the stress in different regions of cells. The displacement field or stress field alone cannot predict the migration ability of cells, as for migration, the contribution of each needs to be taken into account. For softer material, under a constant contractile force and at the specific time, more deformations take place in cells, while at the same time, the force cannot be transmitted fast within cells. However, in stiffer cells, under a specific rate of stress, the body of cells undergoes less deformation, whereas the generated force is transmitted faster to the substrate to overcome moving-resistance forces. To capture both mentioned phenomena, strain energy formulation is used. This energy is the potential energy stored in cells, and for migration, it is transferred to the microenvironment to distort it and help the cell move forward. The strain energy can be calculated by the strain field and stress field. To find the total strain energy ($U$), we need to measure the summation of strain energy over the entire cell [47]. Here, we simplify the energy formulation for two dimensions as follows; however, it can be extended for three-dimensional formulation.

$$U = \frac{1}{2} \iint \sigma(x.y) \cdot \varepsilon(x,y) \mathrm{d}x \mathrm{d}y, \tag{2.2}$$

where $\sigma(x.y)$ is the stress field in $x$ and $y$ coordinates and $\varepsilon(x.y)$ is the strain field in $x$ and $y$ coordinates. Adhesion complexes are connections that integrate cells to their microenvironment (figure 4a). These thin structures have a critical role in migration and clinically are approved to drive cancer cell metastasis [48]. So, the migratory energy can be specifically computed at these adhesion complexes, migratory index, by integrating the dot product of stress vectors with strain vectors over the area of the connections to approximate the migratory capacity of cancer cells

$$\mu = \frac{\iint \varepsilon.\sigma \mathrm{d}A_{\text{adhesion complexes}}}{2A_{\text{adhesion complexes}}}, \tag{2.3}$$

where $A$ is the total area of the connecting regions to the substrate, $\varepsilon$ is the strain field and $\sigma$ is the stress field in connecting regions, and $\mu$ is the energy stored density. This term represents the energy density stored at adhesion complexes, which link cells to the substrate when contractile forces are generated in cells. During the cell movement, this energy is directly transferred to the surrounding microenvironment to distort it. In order to study the migratory index of non-invasive and invasive cells, we assumed that all parameters of cells are identical except the bulk stiffness of cells ($K_1 = 10K_2$, figure 3c). The role of stiffness alone was studied in migration by assessing their migratory index. It should be mentioned that living cells have complex rheological properties due to their complex cross-linked networks and cannot purely be translated in the cell mechanics. However, assuming cells as a pure mechanical system was proven to be sufficient to understand the global behaviour of cells [49].

## 2.4. Strain energy for hyperelastic material

In our simulation, cells (both nucleus and cytoplasm) were assumed as homogeneous and isotropic material. The hyperelastic model can be used to describe the constitutive behaviour of cells [31,49,50]. For isotropic neo-Hookean hyperplastic materials, the strain energy density ($U$) can be expressed by the following equation, and the first variation of this strain energy gives Piola stress [49,50]

$$U = \frac{\mu_o}{2}(\overline{\lambda_1} + \overline{\lambda_2} + \overline{\lambda_3} - 3) + \frac{k_o}{2}(\lambda_1\lambda_2\lambda_3 - 1)^2, \tag{2.4}$$

where $\lambda_i$ are principal stretches, $\overline{\lambda_i}$ are deviatoric principal stretches and $k_o$ and $\mu_o$ are initial bulk and shear moduli, respectively. The initial shear and bulk moduli can be determined by standard linear elastic equations

$$\mu_o = \frac{E}{1 + v} \tag{2.5}$$

and

$$k_o = \frac{E}{3(1 - 2v)}, \tag{2.6}$$

where $E$ is the elastic modulus and $v$ is the Poisson ratio. The cell was assumed to be incompressible material with Poisson's ratio of 0.49 [51], and stress field and displacement field were calculated by

**Table 1.** Material and physical modelling of cells.

| parameters (unit) | values |
| --- | --- |
| elasticity of cytoplasm ($E_{cyto}$) (Pa) | 50–1000 |
| elasticity of nucleus ($E_{nuc}$) | 10 $E_{cyto}$ |
| density of cytoplasm (g cm$^{-3}$) | 1.05 |
| density of nucleus (g cm$^{-3}$) | 1.3 |
| whole cell dimensions (μm) | 3.5 height and 14 length |
| nucleus size—ellipsoidal shape (μm) | 6 major diameter and 2 minor diameter |
| contractile force (Pa) | 5–10 |
| number of connections | 11 |

varying the bulk elasticity of the cells. The deformation of cells and stress in cells can be characterized by the equivalent elastic strain ($\varepsilon^*$) and von Mises stress ($\sigma^*$), respectively [42],

$$\varepsilon^* = \frac{1}{\sqrt{2}(1+v)}[(\varepsilon_{xx} - \varepsilon_{yy})^2 + (\varepsilon_{xx} - \varepsilon_{zz})^2 + (\varepsilon_{yy} - \varepsilon_{zz})^2]^{0.5} \tag{2.7}$$

and

$$\sigma^* = \frac{1}{\sqrt{2}}[(\sigma_{xx} - \sigma_{yy})^2 + (\sigma_{xx} - \sigma_{zz})^2 + (\sigma_{yy} - \sigma_{zz})^2]^{0.5}, \tag{2.8}$$

where $v$ is the Poisson ratio, $\varepsilon_{ii}$ are principal strains in different directions and $\sigma_{ii}$ are principal stresses in different directions. In our simulation, the averages of $\varepsilon^*$ and $\sigma^*$ were computed for different bulk Young's modulus of cells by averaging $\varepsilon^*$ and $\sigma^*$ over the entire relevant area, respectively ($\overline{\varepsilon^*}$ and $\overline{\sigma^*}$).

## 2.5. Numerical simulation

In order to calculate the migratory index for different stiffness values, we developed a two-dimensional finite element model to solve the governing equations (stress field and displacement field). The mechanical module at COMSOL Multiphysics was used to model cells, generate mesh over the entire cell and compute the stored energy by considering the hyperelastic formulation and using above-described physical and material properties. A fully coupled solution method was used to carry out all the simulations and solve the system of equations when the internal contractile forces are applied.

### 2.5.1. Boundary conditions and mesh model

In order to characterize the cellular deformations and force distributions in cells under contractile forces, cells were considered to be fixed on a substrate, as shown in figure 3a. Zero displacements were set at the bottom of the substrate, and free condition was assumed for the top surface of cells. Based on the available data in the literature, to simulate the contractile force, a constant pressure towards the centre of cells in the range of 1–100 Pa [52,53] was applied at the periphery of cells (figure 4a). For covering both normal and cancer cells, the bulk Young's modulus of cells was assumed to vary between 50 and 1000 Pa, and the density of cytoplasm and nucleus were assumed to be 1050 and 1300 kg m$^{-3}$, respectively [54]. The average length of cells and the height of cells were set to be 14 and 3.5 μm. The nucleus for simplicity is assumed to be in ellipsoid shape with a length of the major axis 6 μm and a length of the minor axis 2 μm. All parameters used in the simulation are presented in table 1.

A free triangular mesh was used to generate mesh over the cells (figure 4b). A two-dimensional mesh independency study was performed to choose an efficient mesh type for our simulations. Different mesh sizes were generated (figure 4d), and the $\sigma^*$ on the top surface of the cell for a short distance was computed and plotted in figure 4c to compare their differences. By decreasing the size element and increasing the freedom degree, the differences between results were calculated to reach negligible differences for two successive results. As can be seen in figure 4c, the differences between 'extra fine' and 'extremely fine' mesh types are very small, so 'extra fine' was chosen for an efficient computation run and the following simulation.

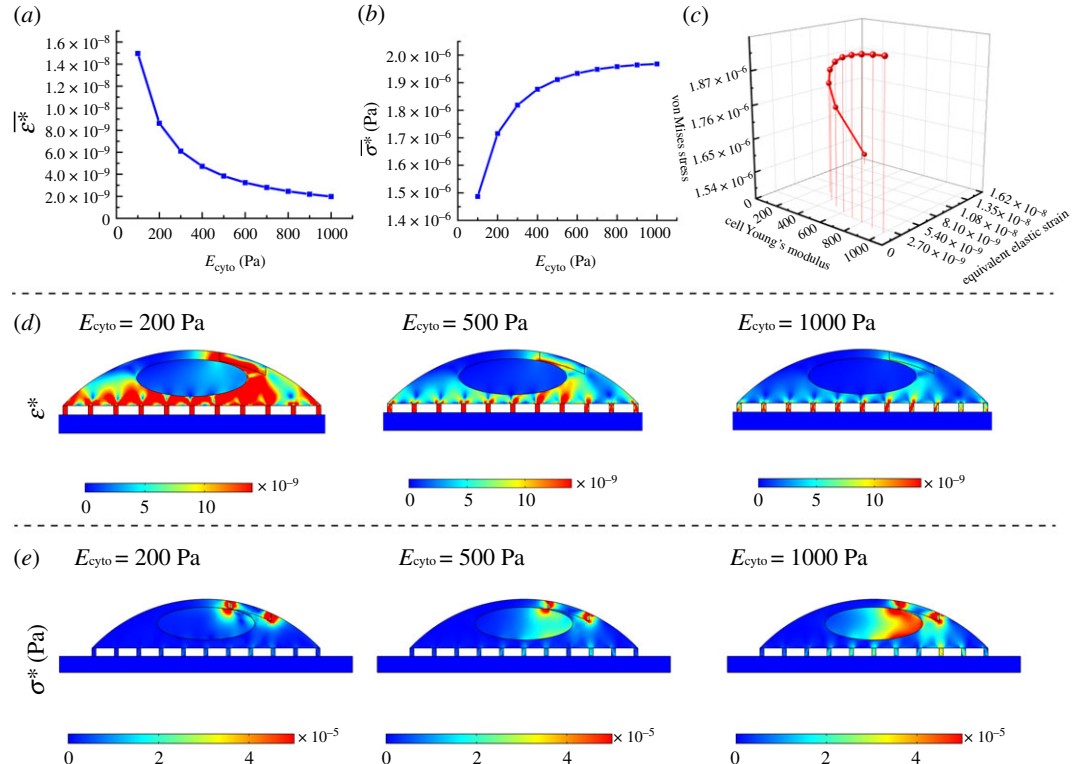

**Figure 5.** Deformation ($\varepsilon^*$) and stress ($\sigma^*$) distribution in cells due to the generated force at actin filaments for different bulk Young's modulus ($E_{cyto}$: 100–1000 Pa). (*a*) Deformation at the cell–substrate interface is decreased by increasing the stiffness of cells, (*b*) stress induced at the cell substrate due to the contractile force for different $E_{cyto}$, (*c*) a three-dimensional plot showing the correlation between $\varepsilon^*$ and $\sigma^*$ for different $E_{cyto}$, (*d*) $\varepsilon^*$ contours by increasing the cell stiffness and (*e*) $\sigma^*$ contours with increasing the stiffness of cell (we assumed the substrate is rigid).

# 3. Results

## 3.1. Cellular deformation and stress

COMSOL Multiphysics® was used to solve the governing equation and quantify the deformation of the whole cells when a constant contractile force is applied at the periphery of cells (rear-middle) [34]. Stress values generated in actin filaments originate from the activity of the molecular motors, myosin II, which are connected to the actin filaments [45,53]. In the following simulations, the physical properties of human breast cancer cells available in the literature were used to numerically calculate the stress within cells and characterize their deformations [42]. Cells with smaller stiffness cannot resist the applied force, so the applied force is reflected in a higher level of deformation. Stiffer cells can propagate the mechanical signal better compared with relatively softer cells, and softer cells can display more deformation compared with the relatively stiffer cells.

The variations of cellular deformation ($\varepsilon^*$) and stress ($\sigma^*$) within cell under the contractile forces (for a short time) can be seen in figure 5*a,b* when the bulk elasticity of cytoplasm ($E_{cyto}$) is varied between 100 and 1000 Pa. At a low $E_{cyto}$, cells undergo a considerable deformation while at a high $E_{cyto}$, cells feel higher stress at their bodies. These different responses are evident in the three-dimensional graph presented in figure 5*c*, where cellular $\varepsilon^*$ and $\sigma^*$ are compared with each other with respect to $E_{cyto}$. Figure 5*d* compares the deformation of the same single cell for three different $E_{cyto}$ (200, 500 and 1000 Pa) once a contractile force is applied in a short time. As can be seen in the surface contours, for very soft cells, a wider surface area of cells undergoes deformation. For relatively stiffer cells, only regions very close to the location of the force are deformed, while at the central region, the nucleus undergoes less deformation. Figure 5*e* shows the $\sigma^*$ distributions and their values for three different Young's modulus. In contrast to $\varepsilon^*$, with increasing cell elasticity, higher values of stress are distributed within cells in the distance from the place of the applied force. Force transmission in cells is determined by the architecture and mechanics of the F-actin networks. These two parameters can be reflected in the bulk stiffness of cells. Therefore, the force transmission level can be estimated by

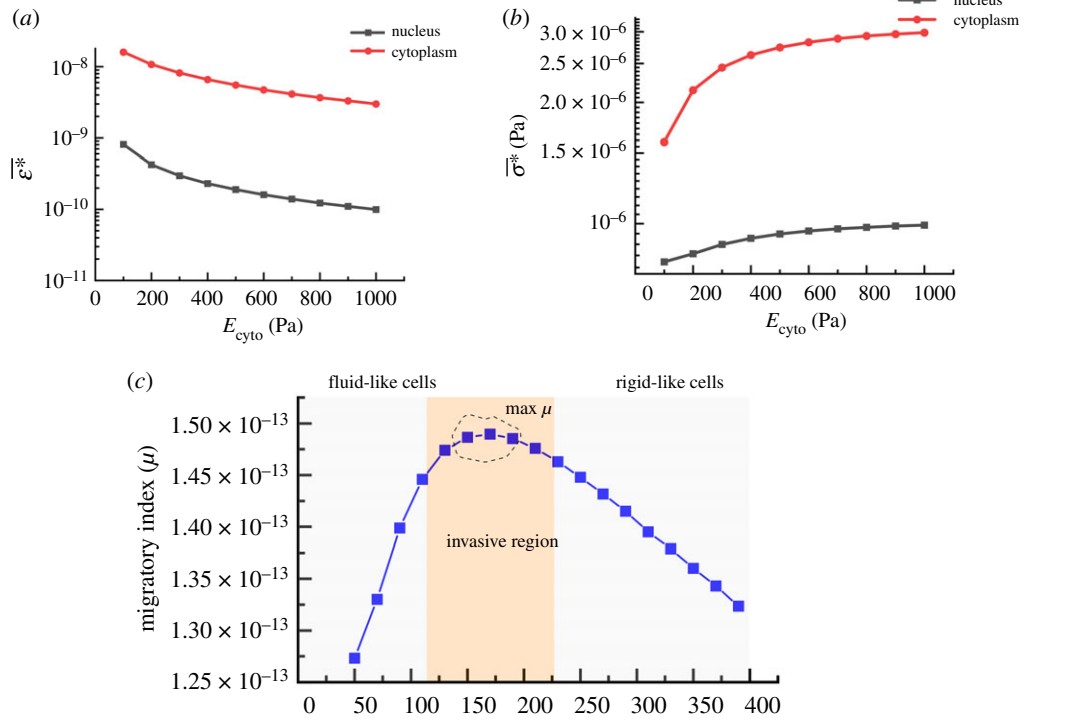

**Figure 6.** (a) Average equivalent elastic strain ($\overline{\varepsilon^*}$) for cytoplasm and nucleus when a contractile force is applied, (b) average stress ($\overline{\sigma^*}$) in the cytoplasm and the nucleus when a contractile force is generated (contractile pressure: 6 Pa, $E_{nuc}$ (elasticity of nucleus) = 10 $E_{cyto}$), (c) migratory index (J m$^{-2}$) quantification of a cell concerning its different bulk Young's modulus when it adheres to a rigid substrate and a contractile force is applied (substrate is completely rigid, contractile pressure: 6 Pa, effective time: 0.001 s).

cell bulk stiffness whose level is influenced by type and quantity of cross-linking proteins. For efficient migration, cells need to transmit the generated force efficiently to the cell–substrate interface at a shorter time. As can be noted in figure 5e, for stiffer cells ($E_{cyto}$ = 1000 Pa) at the same time, the nucleus feels higher values of stress on the left side from the contractile forces generated in the back-middle of cells. The average deformation and the average transmitted stress were quantified separately over cytoplasm and cell nucleus and plotted in figure 6a,b. The nucleus of cells is stiffer compared with other regions of cells, so it undergoes less deformation (one order less) compared with the cytoplasm, and its deformation is reduced by increasing the rigidity of the cells (figure 6a). The average transmitted stress over the cytoplasm also is higher compared with the nucleus; however, its trend is ascending, unlike deformation.

## 3.2. Migratory index ($\mu$)

Neither very stiff nor very soft cells can provide conditions for fast migration, which occurs in the cancer progression. A higher level of deformability disrupts the mechanical signalling within cells, and a higher level of rigidity impedes the cell deformation and, consequently, its movement. It should be a balance between the transmitted stress and deformation in cells to enable them to migrate fast, and any imbalance between these two parameters can suppress the progress of cancer. Therefore, it is necessary to define a unique parameter that covers the contributions of both phenomena. As earlier discussed, with the aid of migratory energy, we could uniquely characterize the migratory capacity of cancer cells. To predict the migratory potential of cells, we need to measure the migratory index for a short time. Once a force is produced in cells, under a stationary condition, when time goes to infinity, regardless of the cell stiffness, the effects of the applied forces will be all transferred to the cell–substrate interface. Hence, with the stationary study, we cannot comprehend the mechanical signalling capacity of the cell. The ability of cells for mechanical signalling needs to be examined by quantifying the transmitted forces for a limited time. So, the mechanical stress values at the cell–substrate interface

(adhesion complexes) need to be quantified by applying contractile forces for a short time. Depending on the stiffness of cells, forces could be transmitted fast or slow.

Figure 6c shows a typical migratory index response of the cell when a contractile force is applied for 1 ms. It is very interesting to observe an optimum range for the cell migratory index when $E_{cyto}$ is varying between 150 and 200 Pa. The peak shows that depending on the physical properties of cells, there is an optimum range of stiffness at which the cell has a maximum potential to migrate on a rigid substrate. It is believed that the optimum range is a critical region as it provides cancer cells with mechanically favourable conditions to enhance their migratory abilities required for cancer progression. By increasing or decreasing the stiffness, as shown in figure 6c, the migratory index of cells or the potential of cells for migration is reduced, indicating that a mismatch occurs between the deformation and the transmitted force. Fluid-like cells with lower stiffness are soft with low capacity to transmit force or deform its environment, while rigid-like cells with high stiffness can deform the environment while undergoing less deformation. Hence, cells with either very low or very high stiffness have low capacity to strain the environment, leading to the low migratory index. Therefore, for a very low and very high stiffness, the migratory capacity of cells is low. Cells for fast migration need to have a higher migratory index. At a specific stiffness/range (maximum migratory index), not only contractile forces sufficiently deform the cells, but also they are transmitted well to the adhesion complexes wherein cells are linked to the substrate. We believe that in cancer, the cytoskeletal organization of cells is changed in a way that the bulk elasticity of cells falls in the critical range at which the migratory index is very high. Based on this approximate model, cells with a higher migratory index could migrate faster under a constant contractile force. It is thought that cancer cells know how to modulate their stiffness to a critical value (stiffness) in which cell mobility is maximum. Estimating this critical stiffness is necessary to understand the behaviour of cancer cells and develop anti-cancer drugs to potentially control the migration of cancer cells, which is essential for cancer progression. The critical stiffness can be influenced by any parameters affecting the cell mechanics such as cell size, cell configuration and external parameters such as substrate stiffness.

## 3.3. Effects of substrate elasticity on the migratory index

Cells can sense their extracellular matrix, and any change in their microenvironment might affect their ability for migration. Substrate elasticity has shown to be an important factor for the migration and invasiveness of cancer cells. Some investigations have proved that the capability of cells for migration is boosted by increasing the substrate rigidity [55]. It has also been substantiated that cells tend to migrate directionally towards places with stiffer substrates (durotaxis) [55]. In order to examine the effects of substrate elasticity computationally, the deformations at connecting regions and transmitted forces within them were quantified by varying the stiffness of the substrate. Figure 7a indicates the average $\varepsilon^*$ when cells are placed on various substrates with different elasticities. As can be seen in the results, cells under a constant contractile force undergo more deformation at their adhesion complexes when the stiffness of the substrate is decreased. However, their differences in $\overline{\varepsilon^*}$ are only prominent at very low substrate stiffness, and for substrate elasticity higher than 5 kPa, their differences in $\overline{\varepsilon^*}$ are almost insignificant. For the transmitted stress ($\sigma^*$), the trend is completely in reverse, by increasing the substrate stiffness, cells at their interface with the substrate sense a higher level of stress. This behaviour can be seen in figure 7b. A stiff substrate resists the local deformation caused by the transmitting force from the internal contractile force, and this resistance is reflected in the form of higher stress at the cell–substrate interface. For a soft substrate, forces internally generated in cells provide more deformations at the substrate and consequently cause less stress at the cell–substrate interface. As already discussed, both deformation and stress at cell–substrate contribute to cell migration and the migratory index. So, any change in either deformation or stress responses of cells (due to the different cell substrates) influences the migratory index, its critical range, as well as the balance between deformation and the stress for optimum migration.

In order to see the effect of the substrate stiffness on the migratory index, we measured the migratory index by changing the substrate stiffness between 0.02 and 20 kPa [56], and plotted the results in figure 7c. It clearly shows that the migratory index values are increased by increasing the substrate stiffness, and for the current cell model, its effects on the migratory index are saturated by increasing substrate elasticity above 10 kPa. So, a stiffer substrate could enhance the mobility potential of cells (migratory index), and allows them to migrate faster. During the solid tumour progression, the extracellular matrix becomes rigid due to increasing the collagen deposition and cross-linking in the tumour stroma [57] and provides a favourable condition for cancer progression by giving them a

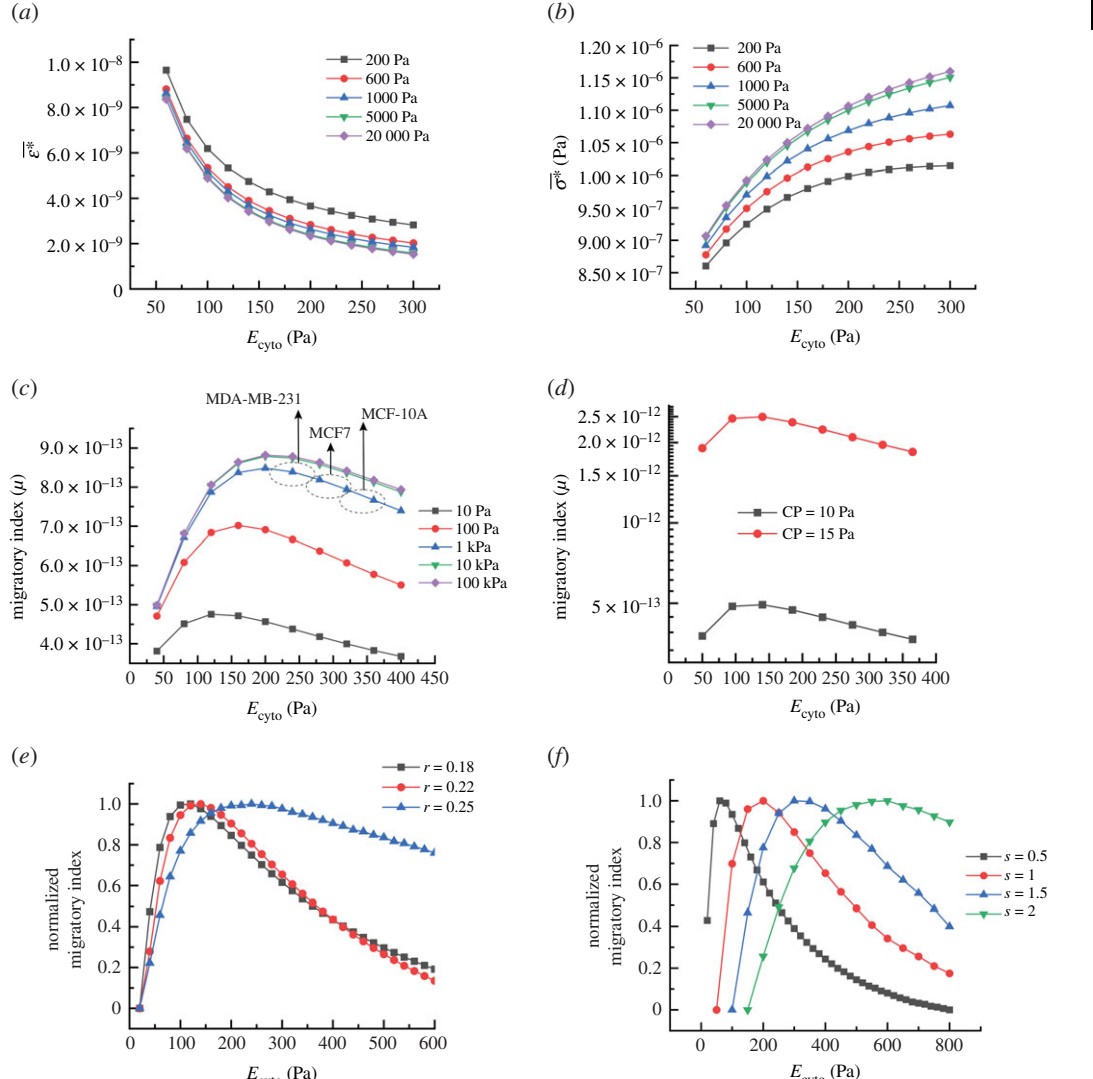

**Figure 7.** (a) Average elastic strain of cell adhesion complexes versus cells' Young's modulus by increasing the elasticity of substrate (between 200 and 20 000 Pa), (b) average stress taking place due to a constant contractile force versus cells' Young's modulus by increasing the substrate elasticity (contractile pressure: 6 Pa, time: 0.001, substrate elastic modulus in Pa), (c) the effects of substrate elasticity on the migratory index (J m$^{-2}$) changes versus cell cytoplasm bulk Young's modulus, (d) migratory index versus cell cytoplasm bulk elasticity increased with the contractile force (CP, contractile force), (e) normalized migratory index showing the effects of aspect ratio on the migratory index peak ($r$ = height/diameter), (f) effects of cells size on the migratory index (s, scale, for $s = 1$: diameter = 14 μm, height = 3.5 μm).

higher mobility potential. Interestingly, it can be seen that the migratory index peak (optimal range) is also shifted by stiffening or softening the substrate, suggesting that the critical stiffness range (peak) could be regulated by altering the mechanical properties of the extracellular matrix as well. In fact, for each substrate stiffness, there is a different critical range of stiffness for cells. These results indicate that on a stiffer substrate, the critical migratory index takes place at a higher stiffness of cells. As shown in figure 7c, for substrate rigidity higher than 5 kPa, no significant change is notable in the migratory index. It seems healthy and non-invasive cells depending on their substrate (microenvironment) stiffness could control their migration by changing their stiffness, and during cancer, cells try to regulate their stiffness to a value at which the migratory index is maximum.

We used the physical parameters of human breast cancer in the current simulations. Here, we can approximate and compare the migratory capability of human breast cancer with different malignancy levels, MCF-10A (benign and non-invasive cancer cells), MCF7 (invasive cancer cells) and MDA-MB-231 (highly invasive cells) by knowing their *in vitro* elasticities [13]. Based on our model (in the case of a stiff substrate), the critical condition for a higher migratory capability is approximated to occur

for cells with elasticity of 200–240 Pa (figure 7$c$) which is in good agreement with the average bulk elasticity of the highly invasive human breast cancer cells (257.5 Pa) previously reported by Corbin *et al*. [13] using a microelectromechanical system (MEMS) resonant sensor. Based on the same study, the average elasticity of the non-invasive human breast cancer cells (MCF-10A) is approximately 398 Pa, which is far from the critical range while reaches down to 257.5 Pa for MDA-MB-231 [13]. The same study showed that the average elasticity of invasive cells (MCF7) is somewhere between two other cells line, but very close to the highly invasive cell lines (275.2 Pa). Based on the migratory model, the migratory potential of highly invasive cells (MDA-MB-231) is in critical range and higher compared with invasive and non-invasive cells. As discussed earlier, invasive cells are assumed to be highly migratory and vice versa [20,23,26–30], suggesting that our model could be also useful to predict the invasiveness of cells based on their bulk stiffness.

## 3.4. Effects of physical parameters on the migratory index

Furthermore, the effects of other parameters on the migratory index were also studied. Internal contractile force directly affects the migratory index level; however, it does not affect the migratory critical position (figure 7$d$). With a higher contractile force, more energy could be stored in cells for migration. The migratory index might be influenced by geometrical factors such as shape, size and aspect ratio. Figure 7$e$ displays the normalized migratory index and its critical range by changing the cells aspect ratio (height/diameter). With increasing cell height, the position of the critical range is shifted to the right, suggesting that for rounder cells, the critical migratory index takes place at a higher cell stiffness compared with the flat cells. The same behaviour can be seen in the migratory index of larger cells (figure 7$f$). The size of cells was increased with the same aspect ratio, and their migratory index values were plotted. The migratory index critical range is shifted to the right by increasing the size of cells, showing that for larger cells, the critical migratory index occurs at a higher stiffness. While not shown here, based on our calculations, the number of connections also could influence the value of the migratory index, but they do not change the optimal ranges.

# 4. Discussion

## 4.1. Migratory index of normal cells and cancer cells

The ability to predict the migration behaviour of cells is necessary to study cancer invasiveness and control cancer progression. Earlier it was discussed and shown that cells depending on their physical characteristics and mechanical properties of their environment exhibit a critical stiffness range for cell movement. This approximate model is based on the fact that cells for migration need to have enough deformation and generate sufficient traction forces to overcome adhesion forces between cells and their extracellular environment [33]. The main driving forces are generated in actomyosin networks and are distributed within cells to drag the body of cells forward [45]. The spatio-temporal force distribution within cells is dependent on the cytoskeletal structures and their mechanical properties such as stiffness. Cells have a complex network of subcellular filaments that allow individual cells to impose mechanical forces on their microenvironment and sustain external forces [46]. These structures of cells and mechanical properties of individual filaments provide a bulk stiffness for each cell. It is believed that in the context of cell mechanics, the migration ability of cells can be approximated by the migratory index, which is dependent on the bulk stiffness of cells. It was shown that for each cell, there is a critical stiffness value/range at which the mobility ability is predicted to be high. This high value is dependent on the size and shape of cells as well. In this range, as shown in the schematic (figure 8), under a constant internal force, not only the contractile force is effectively transmitted through cytoskeletal networks, but also cells have enough deformation, allowing cells to spread fast. Migration plays a central role in cancer progression and metastasis [20,26–29], and based on the migratory index, it is thought that cancerous and invasive cells tend to possess a bulk stiffness in the critical range (between $S1$ and $S2$) at which they have the maximum migratory potential. This critical range is essential to interpret the behaviour of cells based on their cell mechanics; however, further detailed computational and experimental studies are required to determine the extent of the critical migratory index ($S1$ and $S2$).

Many healthy cells (or non-invasive cells) were experimentally proved to have stiffer bodies compared with their counterparts'. Based on our approximate model, they are thought to be at the right side of the critical range with a low migratory index, as shown in figure 8. Cells have dynamic

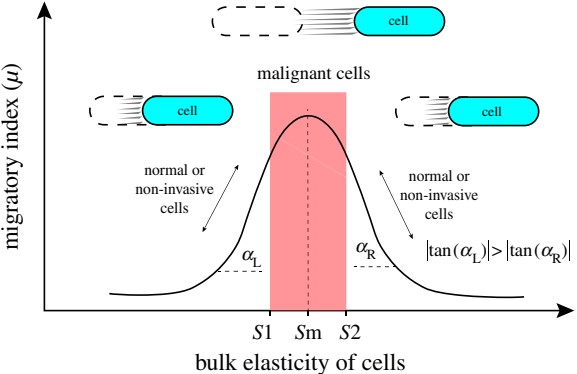

**Figure 8.** Typical migratory index responses of cells and their changes by altering cell stiffness. At the critical value/range (red region), the migratory index is maximum, and cells could move fast; outside of the critical range, cell movement is reduced either by reducing cells stiffness or enhancing the stiffness of the cell.

structures that can be softened or stiffened in response to external cues [31]. This nature of cells enables them to control their migration by changing mechanical properties. Based on our approximate model, as can be seen in figure 8, the migration ability of cells (migratory index) can be regulated by a small variation in bulk stiffness of normal cells. It is thought that normal and non-invasive cells based on their permanent local microenvironment tend to have a stable and tuneable migratory index, which is far enough from the critical range. By a small change in their stiffness, once is necessary, they can either reduce their migration ability or accelerate their migration ability. Based on the migratory model, normal cells (or non-invasive cells) can even have smaller elasticity versus invasive cells (left side of the critical region), and for a fast migration, cells need to become stiff to have a higher migratory index. As discussed in the Introduction, a few cancerous and invasive cells were proved to be stiffer compared with healthy and non-invasive cells [18,19,34]. The migratory model suggests that all invasive cancer cells are not necessarily soft compared with non-invasive cells.

## 4.2. Normal cells have tuneable migratory index

Moreover, as was observed in figure 6c (based on the approximate model), the pace of changes in the migratory index at the right side of the critical range is slower (smaller slope) compared with the left side, providing the normal and non-invasive cells with more control on their migrations. However, the rate of the migratory index at the left side of the critical range is higher (higher slope), which reduces the control resolution on the migratory index. These behavioural changes of the migratory index could interestingly express the reason why many normal cells are stiffer compared with cancer cells and tend to be on the right side. In contrast with the normal cells (or non-invasive cells), in invasive or metastatic cells, the bulk stiffness is reduced until reaching the critical range (the broad peak point), and at that point, any small variation in the bulk stiffness will not affect the movement capacity of cells significantly. Therefore, it is predicted that cancer cells with critical stiffness have less control over their movement abilities, and they lose their ability to adjust their potentials for migrations. At the critical range, small variations in the bulk stiffness cannot be effective in controlling the mobility potential of cells.

## 4.3. Migratory index for developing anti-cancer drugs

Based on the approximate model, the migratory potential of cells can be controlled through increasing or decreasing the bulk stiffness to a value beyond the critical range. Targeting the cytoskeletal structures and altering the mechanical properties of cells can be a useful strategy to manage cell migration required for cancer progression. This process can be caused to occur by designing anti-cancer drugs or nanoparticles-based drugs to disrupt the cytoskeletal organization of cells to reduce the stiffness of cells. Or, through enhancing cell stiffness by stimulating the overexpression of proteins which are responsible for cell mechanics. Both ways can be effective to suppress migratory levels of malignant cells by modulating the mechanobiological properties of cells [23,24]. Based on the migratory index model, measuring the elasticity of both treated and non-treated cells allows us to approximate the efficacy of anti-cancer drugs designed to control cancer migration. The migratory index model can be implemented to

develop effective drugs to control cancer migration. By measuring the migratory index of cells when they are treated with anti-cancer drugs, we can monitor whether anti-cancer drugs are effective or not. Then, we can optimize the effectiveness of anti-cancer drugs by either increasing the cell elasticity from $Sm$ to $S2$ or decreasing the elasticity from $Sm$ to $S1$ to control cancer migration required for cancer progression (figure 8). To inhibit the migratory ability of cancer cells, we need to design anti-cancer drugs with anti-elastic properties in order to reduce their migratory ability. In the context of cell mechanics, an anti-elastic drug will help us to alter the elasticity of cancer cells such that the migratory index is reduced. In the literature, some investigations have proved that anti-cancer drugs or nanoparticles can be used to control the cancer migration and invasiveness by either softening or stiffening their elasticities [23,24,58,59].

# 5. Conclusion

In this study, we introduced our 'migratory index' on the migration ability of cells based on their mechanical proprieties, particularly cell stiffness, to find out the approximate correlation between cancer migration and their deformability levels. In our study, to approximate the mobility potential of cells, we assumed that the initial bulk stiffness of cells and the adhesion force remained constant during migration, and the internal force is independent from the bulk stiffness of cells. Cell migration is essential for cancer cells to be invasive, and it is thought that the migratory index could provide a platform to approximate the invasive level of cancer. The migratory index is defined based on the bulk stiffness of cells and stored strain energy in cells when internal contractile forces are generated. Our mechanical modelling and computational studies showed that non-invasive cells (or normal cells) should not necessarily be stiffer than invasive cells (or cancer cells). However, our results showed that rigid-like normal cells have more control over their movements compared with fluid-like normal cells. Based on the suggested model, stiffness of invasive cells provides them a critical migratory index that allows them to migrate fast and spread to other parts of the body. Based on the migratory index model, during the cancer progression, cells can become either soft or stiff, depending on their normal conditions in the migratory index profile. Our model can also predict the contradictory responses of anti-cancer drugs targeting cell mechanics. From the mechanics point of view, the migratory levels of cells can be controlled by designing drugs to either stiffen or soften cells. However, the level of the required changes needs further detailed study. In this work, we developed a simplified two-dimensional model to show the potential of this concept to predict the migratory potential of cells based on their bulk stiffness. However, in the two-dimensional model, we did not simulate all parameters existing *in vivo* environment such as other forces influencing cells. For future work, a three-dimensional model can be developed to measure migratory index in a more realistic environment while other parameters are considered. Our migratory index model provides valuable information to understand why the stiffness of cells is increased or decreased during cancer progression and helps to take different strategies to control migration of cancer cells potentially.

Data accessibility. All data exported from the COMSOL Multiphysics. The CAD model, setting parameters for COMOSL simulation and exported data for few figures can be found from the Dryad Digital Repository: https://dx.doi.org/10.5061/dryad.ns1rn8ppd [60]. The other data generated during and/or analysed during the current study are available from the corresponding author on reasonable request.

Authors' contributions. M.P. and A.S.K. conceived and designed the study. A.S.K. carried out the simulations, analysed the data and drafted the manuscript. M.P. supervised the study and both authors reviewed the manuscript and gave approval for publication.

Competing interests. The authors declare no competing interests.

Funding. Natural Sciences and Engineering Research Council of Canada (NSERC), Concordia Research Chair (CURC), and Fonds Québécois de la Recherche sur la Nature et les Technologies (FRQNT) grants to M.P., and doctoral FRQNT to A.S.K. are acknowledged.

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
