## [Reviewer comments · Royal Society Open Science]

Review History

RSOS-200747.R0 (Original submission)

Review form: Reviewer 1

Is the manuscript scientifically sound in its present form?

No

Are the interpretations and conclusions justified by the results?

No

Is the language acceptable?

Yes

Do you have any ethical concerns with this paper?

No

Have you any concerns about statistical analyses in this paper?

No

Recommendation?

Major revision is needed (please make suggestions in comments)

Comments to the Author(s)

1. The manuscript is concerned with cancer cells optimize elasticity for efficient migration, which is interesting. It is relevant and within the scope of the journal.
2. However, the manuscript, in its present form, contains several weaknesses. Adequate revisions to the following points should be undertaken in order to justify recommendation for publication.
3. My major objection concerns this issue that I could not see any validation or assessment for the model. The proposed 2D model needs to be assessed via in vitro observations.
 - There are also some minor comments:
4. The Introduction section is too long. I recommend to add an appendix and put some of the general material there.
5. Equation 1 need proper reference.
6. For readers to quickly catch the contribution in this work, this would be great if the authors briefly mention the results they have obtained from the analyses here.

Review form: Reviewer 2

Is the manuscript scientifically sound in its present form?

No

Are the interpretations and conclusions justified by the results?

Yes

Is the language acceptable?

Yes

Do you have any ethical concerns with this paper?

No

Have you any concerns about statistical analyses in this paper?

No

Recommendation?

Major revision is needed (please make suggestions in comments)

Comments to the Author(s)

In this study, the authors presented a numerical model of cellular deformation considering several affecting parameters such as cell elasticity, cell size, and substrate stiffness. They introduced migratory index to show how elastic modulus of cancer/ normal cells can be related to their migratory behavior. They presented a 2D model of cells considering different elastic properties as the representors of healthy and normal cell and concluded that the migration potential of cells is defined based on the stored energy. They emphasized that both cell deformability and force transmission affect the stored energy and consequently the migration potential of cells. Although the topic of this manuscript is interesting, however, there are several major and critical issues which need to be addressed carefully.

Major:

- The manuscript does not have a suitable structure as a journal paper. It should be clearly divided into sub-sections starting from introduction, then method, result or result/discussion, and conclusion. The authors are expected to re-organize different parts of the manuscript to fit a research paper requirements for example:

- o The introduction section is very long with a lot of non-related information. It seems more like a chapter book or review paper and needs to be re-organized. There is no need to define some simple parameters such as deformation. Also, page 7, the effect of different drugs can be summarized as it is not related to what has been done in this work.

- o The section numbering needs to be corrected starting from 1. Introduction. It supposes to put all method sections together before talking about results. I suggest to merge all method sections together before starting results (e.g. section 1.3.2 migratory index. The method of calculation and definition should go to the method section).

- It is not clear whether Figure 2 has been taken from another work or not? If it is generated by authors through literature review, I suggest to move it from introduction to the method. Also. Please provide more information on the method of stiffness measurement for each study. There are different methods to measure cellular stiffness which could measure either local stiffness (such as AFM), or bulk stiffness (such as micropipette aspiration).

- I think one of the main limitations of current study is considering a single cell instead of a population of cells which play an important role in cellular migration, force transmission, and deformability. I strongly suggest to include at least two cells, considering their cell-cell interactions (similar to cell-surface interactions that authors have already considered).

- Authors need to clarify how defined cell-substrate interactions. How many connections were considered, do you think the number of connections will affect the results?

- A huge number of studies have demonstrated different behavior of cells in 2D and 3D environment. While considering 3D model still can keep the model simple and efficient, why authors decided to choose 2D model?

- Page 27- it has been mentioned that “the stiffness of cancer cells was assumed (277.3 Pa) based on previously reported by Corbin et al. [59] by AFM measurements”. Did authors note that AFM measures local stiffness (surface of the cell) not the bulk stiffness? While the provided model is based on the bulk stiffness why authors considered local stiffness measured by AFM instead of using bulk stiffness of cells which has been reported in several papers.

- Authors examined the effect of substrate stiffness, but such study can be useful when the substrate is considered as a deformable material. The migration behavior of cells strongly depends on substrate stiffness and its deformation. How authors address this point?

Minor:

- Page 8, defining the rigidity index requires a reference.

- The section numbering need to be modified starting from 1 for introduction, then 2. Method (including 2.1, 2.2, etc.)

- There is no need to define cytoplasm in page 11

- I suggest to add one table summarize all input parameters

- Some figures can be merged such as figure 6 and 7 together or figure 8 and 9 together.

- Section 1.6 Migratory Index Model for Developing Anti-Cancer Drugs, is a part of discussion and does not have separate results, so there is no need to have a separate numbering for that.

Review form: Reviewer 3

Is the manuscript scientifically sound in its present form?

Yes

Are the interpretations and conclusions justified by the results?

Yes

Is the language acceptable?

Yes

Do you have any ethical concerns with this paper?

No

Have you any concerns about statistical analyses in this paper?

No

Recommendation?

Accept with minor revision (please list in comments)

Comments to the Author(s)

In this manuscript, these authors defined a quite interesting energy-depend term, a migratory index which could indicate cell migration potential, to explain the contradiction about whether the speed of cell migration and invasion were inversely correlated to cell stiffness. Their mechanical model showed that cells had a maximum migratory index taking place at a specific range of cell bulk elasticity by control cell physical parameters, indicating the maximum capability of cell migration. The authors also offered valuable advice about adopting different cell stiffness adjustment strategies to control the migration of cancer cells. Nevertheless, they needed more information to explain some terms and the Migration Index formula. More specific comments are detailed below:

1. The authors needed more explanation in “non-invasive counterparts” in the section of abstract “We believe that the stiffness of invasive cells depending on the stiffness of their non-invasive counterparts is either decreased or increased to reach the critical condition in which the mobility potential of cells is approximated to be maximum.” and introduction “Where r is the rigidity index, E_c is the elasticity of cancer or invasive cells, and E_n is the elasticity of the normal or non-invasive counterparts with the same unit.” What did non-invasive counterparts mean? Was it the substrate or surrounding cells? Or was it normal cells before cell became more invasive? If non-invasive counterparts meant normal cells, how to explain the invasive and non-invasive cells in human kidney, bladder, and thyroid cell in fig 1?
2. In the section of the introduction, “Beyond that range, cells might lose their capability to migrate fast and potentially move into other parts of the body.” If cells lost their capability to migrate fast, how could they move into other parts of the body?
3. In the section of “1.21 Cell Modeling and Force Generation”, Was it contradictory to assume that the bulk stiffness was constant and independent of contraction force and that cells were considered as a pure elastic mechanical body with a bulk stiffness?
4. In the section of “1.32 Migratory Index(μ)”, the authors defined a formula for calculation the migratory index (μ), what did the Adhesion complexes mean? At the same time, the authors needed more explain the form of this formula.
5. In the section of “1.33 Effects of substrate elasticity on the migratory index” and Fig 9, the authors needed a better explanation that the capability of cells for migration is boosted by increasing the substrate rigidity. Was it related to the value of μ or the peak position? If cell migration related to the value of μ , was the migration capacity of MDA-MB-231 cells on a 10 Pa substrate less than the migration capacity of MCF-10A on 100 kPa?

Decision letter (RSOS-200747.R0)

Dear Dr Packirisamy,

The editors assigned to your paper ("Cancer Cells Optimize Elasticity for Efficient Migration") have now received comments from reviewers. We would like you to revise your paper in accordance with the referee and Associate Editor suggestions which can be found below (not including confidential reports to the Editor). Please note this decision does not guarantee eventual acceptance.

Please submit a copy of your revised paper before 28-Aug-2020. Please note that the revision deadline will expire at 00.00am on this date. If we do not hear from you within this time then it will be assumed that the paper has been withdrawn. In exceptional circumstances, extensions may be possible if agreed with the Editorial Office in advance. We do not allow multiple rounds of revision so we urge you to make every effort to fully address all of the comments at this stage. If deemed necessary by the Editors, your manuscript will be sent back to one or more of the original reviewers for assessment. If the original reviewers are not available, we may invite new reviewers.

- Data accessibility

If you wish to submit your supporting data or code to Dryad (<http://datadryad.org/>), or modify your current submission to dryad, please use the following link:
<http://datadryad.org/submit?journalID=RSOS&manu=RSOS-200747>

- **Competing interests**

- **Authors' contributions**

- **Acknowledgements**

- **Funding statement**

on behalf of Dr Derek Abbott (Associate Editor) and Pietro Cicuta (Subject Editor)
openscience@royalsociety.org

Comments to Author:

Reviewers' Comments to Author:

Reviewer: 1

Comments to the Author(s)

1. The manuscript is concerned with cancer cells optimize elasticity for efficient migration, which is interesting. It is relevant and within the scope of the journal.

2. However, the manuscript, in its present form, contains several weaknesses. Adequate revisions to the following points should be undertaken in order to justify recommendation for publication.
3. My major objection concerns this issue that I could not see any validation or assessment for the model. The proposed 2D model needs to be assessed via in vitro observations.
 - There are also some minor comments:
4. The Introduction section is too long. I recommend to add an appendix and put some of the general material there.
5. Equation 1 need proper reference.
6. For readers to quickly catch the contribution in this work, this would be great if the authors briefly mention the results they have obtained from the analyses here.

Reviewer: 2

Comments to the Author(s)

In this study, the authors presented a numerical model of cellular deformation considering several affecting parameters such as cell elasticity, cell size, and substrate stiffness. They introduced migratory index to show how elastic modulus of cancer/ normal cells can be related to their migratory behavior. They presented a 2D model of cells considering different elastic properties as the representors of healthy and normal cell and concluded that the migration potential of cells is defined based on the stored energy. They emphasized that both cell deformability and force transmission affect the stored energy and consequently the migration potential of cells. Although the topic of this manuscript is interesting, however, there are several major and critical issues which need to be addressed carefully.

Major:

- The manuscript does not have a suitable structure as a journal paper. It should be clearly divided into sub-sections starting from introduction, then method, result or result/discussion, and conclusion. The authors are expected to re-organize different parts of the manuscript to fit a research paper requirements for example:
 - o The introduction section is very long with a lot of non-related information. It seems more like a chapter book or review paper and needs to be re-organized. There is no need to define some simple parameters such as deformation. Also, page 7, the effect of different drugs can be summarized as it is not related to what has been done in this work.
 - o The section numbering needs to be corrected starting from 1. Introduction. It supposes to put all method sections together before talking about results. I suggest to merge all method sections together before starting results (e.g. section 1.3.2 migratory index. The method of calculation and definition should go to the method section).
- It is not clear whether Figure 2 has been taken from another work or not? If it is generated by authors through literature review, I suggest to move it from introduction to the method. Also. Please provide more information on the method of stiffness measurement for each study. There are different methods to measure cellular stiffness which could measure either local stiffness (such as AFM), or bulk stiffness (such as micropipette aspiration).
- I think one of the main limitations of current study is considering a single cell instead of a population of cells which play an important role in cellular migration, force transmission, and deformability. I strongly suggest to include at least two cells, considering their cell-cell interactions (similar to cell-surface interactions that authors have already considered).
- Authors need to clarify how defined cell-substrate interactions. How many connections were considered, do you think the number of connections will affect the results?

- A huge number of studies have demonstrated different behavior of cells in 2D and 3D environment. While considering 3D model still can keep the model simple and efficient, why authors decided to choose 2D model?
- Page 27- it has been mentioned that “the stiffness of cancer cells was assumed (277.3 Pa) based on previously reported by Corbin et al. [59] by AFM measurements”. Did authors note that AFM measures local stiffness (surface of the cell) not the bulk stiffness? While the provided model is based on the bulk stiffness why authors considered local stiffness measured by AFM instead of using bulk stiffness of cells which has been reported in several papers.
- Authors examined the effect of substrate stiffness, but such study can be useful when the substrate is considered as a deformable material. The migration behavior of cells strongly depends on substrate stiffness and its deformation. How authors address this point?

Minor:

- Page 8, defining the rigidity index requires a reference.
- The section numbering need to be modified starting from 1 for introduction, then 2. Method (including 2.1, 2.2, etc.)
- There is no need to define cytoplasm in page 11
- I suggest to add one table summarize all input parameters
- Some figures can be merged such as figure 6 and 7 together or figure 8 and 9 together.
- Section 1.6 Migratory Index Model for Developing Anti-Cancer Drugs, is a part of discussion and does not have separate results, so there is no need to have a separate numbering for that.

Reviewer: 3

Comments to the Author(s)

In this manuscript, these authors defined a quite interesting energy-depend term, a migratory index which could indicate cell migration potential, to explain the contradiction about whether the speed of cell migration and invasion were inversely correlated to cell stiffness. Their mechanical model showed that cells had a maximum migratory index taking place at a specific range of cell bulk elasticity by control cell physical parameters, indicating the maximum capability of cell migration. The authors also offered valuable advice about adopting different cell stiffness adjustment strategies to control the migration of cancer cells. Nevertheless, they needed more information to explain some terms and the Migration Index formula.

More specific comments are detailed below:

1. The authors needed more explanation in “non-invasive counterparts” in the section of abstract “We believe that the stiffness of invasive cells depending on the stiffness of their non-invasive counterparts is either decreased or increased to reach the critical condition in which the mobility potential of cells is approximated to be maximum.” and introduction “Where r is the rigidity index, E_c is the elasticity of cancer or invasive cells, and E_n is the elasticity of the normal or non-invasive counterparts with the same unit.” What did non-invasive counterparts mean? Was it the substrate or surrounding cells? Or was it normal cells before cell became more invasive? If non-invasive counterparts meant normal cells, how to explain the invasive and non-invasive cells in human kidney, bladder, and thyroid cell in fig 1?
2. In the section of the introduction, “Beyond that range, cells might lose their capability to migrate fast and potentially move into other parts of the body.” If cells lost their capability to migrate fast, how could they move into other parts of the body?
3. In the section of “1.21 Cell Modeling and Force Generation”, Was it contradictory to assume that the bulk stiffness was constant and independent of contraction force and that cells were considered as a pure elastic mechanical body with a bulk stiffness?
4. In the section of “1.32 Migratory Index(μ)”, the authors defined a formula for calculation the migratory index (μ), what did the Adhesion complexes mean? At the same time, the authors needed more explain the form of this formula.

5. In the section of “1.33 Effects of substrate elasticity on the migratory index” and Fig 9, the authors needed a better explanation that the capability of cells for migration is boosted by increasing the substrate rigidity. Was it related to the value of μ or the peak position? If cell migration related to the value of μ , was the migration capacity of MDA-MB-231 cells on a 10 Pa substrate less than the migration capacity of MCF-10A on 100 kPa?

Author's Response to Decision Letter for (RSOS-200747.R0)

See Appendix A.

Decision letter (RSOS-200747.R1)

Dear Dr Packirisamy,

It is a pleasure to accept your manuscript entitled "Cancer Cells Optimize Elasticity for Efficient Migration" in its current form for publication in Royal Society Open Science.

on behalf of Dr Derek Abbott (Associate Editor) and Pietro Cicuta (Subject Editor)
openscience@royalsociety.org

Appendix A

Response to reviewers regarding the manuscript entitled:

Cancer Cells Optimize Elasticity for Efficient Migration by *Ahmad Sohrabi Kashani, and Muthukumaran Packirisamy*

Dear Editor,

We would like to thank the reviewers for a careful and thorough reading of our manuscript and their thoughtful comments and constructive suggestions. We have addressed all their comments and suggestions and revised our manuscript. In the following, please find our detailed responses:

Reviewer 1 (R1)

Reviewer 2 (R2)

Reviewer 3 (R3)

Purple-marked: moved parts/sections

Highlighted: Revised parts

Reviewer 1

Major Comments:

The manuscript is concerned with cancer cells optimize elasticity for efficient migration, which is interesting. It is relevant and within the scope of the journal.

Thanks for your positive feedback on the manuscript

However, the manuscript, in its present form, contains several weaknesses. Adequate revisions to the following points should be undertaken in order to justify recommendation for publication.

In the following parts, please find our detailed responses.

R1- Comments #1: My major objection concerns this issue that I could not see any validation or assessment for the model. The proposed 2D model needs to be assessed via *in vitro* observations.

Thanks for pointing out this concern. In this manuscript, we only developed a theoretical model and approach to introduce “migratory index” concept in order to understand the influence of the bulk stiffness on migration. Many studies (cited in the manuscript) experimentally studied *in vitro* the migration ability of cells and their mechanical properties. Even though the results seem not converging, our migratory index model is in good agreement with their reported results and providing a way of understanding the results in a more comprehensive way. In a case of human breast cancer cell lines, for example, highly invasive cells (MDA-MB-231), our approximate model predicts well when the elasticity is largely reduced, the migration ability of cells is increased and “migratory index” reaches to the peak point. In order to verify our model, in **Figure 7c**, we compared the prediction of our model with more details with mechanical properties measured by experimental methods on human breast cancer cells. In this manuscript, we used the physical properties of human breast cancer cells and as described there, our model is able to predict the experimental observations.

Minor Comments

There are also some minor comments:

- The Introduction section is too long. I recommend to add an appendix and put some of the general material there.

As per your suggestion and Reviewer 2, we removed some parts of the introduction and moved some other parts (Figure 1) to “method” section to make the introduction shorter.

- Equation 1 need proper reference.

We did not take the equation from any reference and defined equation 1 to compare the elasticity of cancer cells and normal cells. We wanted to discuss increase and decrease in stiffness when cells become cancerous.

Action taken:

We corrected the relating part as follow:

Page 9: “For cancer or ~~invasive~~-malignant cells, the rigidity index could be defined with respect to the elasticity of their normal or ~~non-invasive-malignant~~ counterpart cells as follow:”

- For readers to quickly catch the contribution in this work, this would be great if the authors briefly mention the results they have obtained from the analyses here.

Thanks for your suggestion. Briefly, in this work, we introduced a new concept/model called “migratory index” to approximate the relationship between cellular migration and elasticity of cells when the cell body is considered as pure mechanical structures. Both increasing and decreasing of the stiffness of cells and the resulting increase in cell migration have been reported in literature. Yet, it is not fully explored how stiffness would contribute to the migration capability of cells. In this work, we suggested that energy-based “migratory index” is able to predict the cell migration capability based on their bulk stiffness. Migratory index, uniquely, is a single term that captures two parameters affecting cell migration, cell deformation, and force transmission, to predict the migratory potential of cells. Our method revealed an optimum range of stiffness in which the migratory index is predicted to be maximum and we believe this is the favorable range for cancer cells to facilitate their migrations. This term could be a function of physical properties such as size and the mechanical properties of their microenvironment.

Reviewer 2

Comments to the Author(s)

In this study, the authors presented a numerical model of cellular deformation considering several affecting parameters such as cell elasticity, cell size, and substrate stiffness. They introduced migratory index to show how elastic modulus of cancer/ normal cells can be related to their migratory behavior. They presented a 2D model of cells considering different elastic properties as the representors of healthy and normal cell and concluded that the migration potential of cells is defined based on the stored energy. They emphasized that both cell deformability and force transmission affect the stored energy and consequently the migration potential of cells. Although the topic of this manuscript is interesting, however, there are several major and critical issues which need to be addressed carefully.

Major Comments

R2- Comment #1: The manuscript does not have a suitable structure as a journal paper. It should be clearly divided into sub-sections starting from introduction, then method, result or result/discussion, and conclusion. The authors are expected to re-organize different parts of the manuscript to fit a research paper requirements for example:

Thanks for your comment on the arrangement of the manuscript.

Action taken:

We revised the manuscript according to the remark. We shortened the introduction, removed parts of the introduction, moved Figure 1 to “method” section, and added a separate part for “discussion”.

R2- Comment #2: The introduction section is very long with a lot of non-related information. It seems more like a chapter book or review paper and needs to be re-organized. There is no need to define some simple parameters such as deformation. Also, page 7, the effect of different drugs can be summarized as it is not related to what has been done in this work.

As per your suggestion, we summarized the paragraph relating to the effects of anti-cancer drugs and removed the other parts to make the introduction shorter. With discussing the anti-cancer drugs, we wanted to emphasize the contradictory results on increasing or decreasing the stiffness with anti-cancer drugs and migration control. Our model could find its applications in designing drugs to alter mechanical properties in a way to reduce the migratory index of cells. We thought of defining a few fundamental engineering and biological parameters involved in migratory index to help both non-engineering and engineering readers appreciate the model better.

Action taken:

We removed most data presented on page 6 relating to anti-cancer drugs. We removed the definition of deformation.

R2- Comment #3: The section numbering needs to be corrected starting from 1. Introduction. It supposes to put all method sections together before talking about results. I suggest to merge all method sections together before starting results (e.g. section 1.3.2 migratory index. The method of calculation and definition should go to the method section).

We changed the numbering according to this remark.

Action taken:

We changed the numbering and merged “migratory index” and associated explanations into the “method” section. (Page 13 and 14)

R2- Comment #4: It is not clear whether Figure 2 has been taken from another work or not? If it is generated by authors through literature review, I suggest to move it from introduction to the method. Also. Please provide more information on the method of stiffness measurement for each study. There are different methods to measure cellular stiffness which could measure either local stiffness (such as AFM), or bulk stiffness (such as micropipette aspiration).

We generated that figure based on the available data in the literature on the elasticity of cancer and normal cells. We discussed two scenarios that might happen during cancer; increasing and decreasing the stiffness and that figure supports our discussion.

Action taken:

We moved that part including rigidity index to the “method” section according to this comment. In the caption of **Figure 2**, we mentioned that cells have been measured with different methods, AFM and micropipette aspiration. All of them used AFM and only the work with reference [36] used micropipette aspiration.

Page 10: “Cells stiffness values have been measured with different methods, Stiffer: AFM [17][18][35], Micropipette aspiration [36], Softer: AFM [3][9][12][14][37][38][39][40][41][42]”

R2- Comment #5: I think one of the main limitations of current study is considering a single cell instead of a population of cells which play an important role in cellular migration, force transmission, and deformability. I strongly suggest to include at least two cells, considering their cell-cell interactions (similar to cell-surface interactions that authors have already considered).

Thanks for pointing out this concern. Cell-cell interaction might affect the migration of cells particularly in collective migration. But, we should mention that we studied the migration ability of cells when they undergo epithelial-mesenchymal transition (EMT) where their transcriptional program is changed and leads downregulations of the cell-cell adhesion. Moreover, as we explained in the manuscript, we intended to investigate how bulk stiffness of cells affects their migratory potential of individual cells when a contractile force is generated. As we discussed in the conclusion, external parameters might alter the migratory potential of cells such as their microenvironment. But, the proposed model can incorporate cell-cell interactions with more research done. We believe cell-cell interaction could be the focus of a separate work and we would need a further detailed study.

Related references:

Bocci, F. et al. Toward understanding cancer stem cell heterogeneity in the tumor microenvironment. *Proc. Natl. Acad. Sci. U. S. A.* 116, 148–157 (2019).

Chaffer, C. L. & Weinberg, R. A. A perspective on cancer cell metastasis. *Science* (80-.). 331, 1559–1564 (2011).

Action taken:

In the introduction, we emphasized that our modeling is studying the migratory potential during the epithelial-mesenchymal transition.

Page 8 : “...the migratory ability of cells during epithelial-mesenchymal transition[10] ...”

R2- Comment #6: Authors need to clarify how defined cell-substrate interactions. How many connections were considered, do you think the number of connections will affect the results?

We assumed that cells are connected to their microenvironments through a series of adhesion complexes. These adhesion complexes integrate the extracellular matrix with the actin cytoskeleton. In our calculation, we assumed that each cell adhered to the substrate with 11 connections. The dimension and size of these connections are presented in the uploaded CAD file. Based on our results and calculations, the number of connections is not going to affect the optimal range; however, they could increase the migratory index. More connections, facilitating greater potential to store energy, and hence higher forces could be transmitted to the substrate. Here, we emphasized the optimal range of stiffness which helps cells speed up their migration on a substrate with a constant elasticity. This study shows how stiffness itself contributes to the migration of cells.

Action taken:

Table 1 was added and the number of connections was presented there. An explanation was also added at the end of the section 1.3.4 to clarify that:

Page 27: “While not shown here, based on our calculations, the number of connections also could influence the value of the migratory index, but they don’t change the optimal ranges.”

R2- Comment #7: A huge number of studies have demonstrated different behavior of cells in 2D and 3D environment. While considering 3D model still can keep the model simple and efficient, why authors decided to choose 2D model?

Here, in this work, we used a 2D model to introduce a new concept, migratory index, to predict the migratory behavior of cells. Although 3D modeling could provide more realistic results, many 2D simulations have been used to predict the behavior of cancer cells [John et al, A Review of Cell-Based Computational Modeling in Cancer Biology, 2019, JCO]. Using the 2D migratory index, we were able to confirm the existence of the optimal range. We believe in this work, the 3D model is not going to provide more details on the migratory concept/model. In terms of bulk stiffness, our 2D model and results are encouraging when they are compared with the experimental data reported in the literature, and we were able to predict their behaviors. Moreover, for human breast cancer cells and for the reported mechanical results which we considered as a case study, the detailed 3D modeling was not available. The 3D model is more essential when the cell migrations need to be considered in a 3D microenvironment where cells may contact their microenvironments from different sides while in the current work we focused on 2D cell migration when cells are only in contact with the substrate. Many migration and invasion assays discussed in this manuscript have been done under the 2D condition where cells are migrating on a flat substrate. As we suggested in the conclusion as to future work, 3D modeling can be used to measure the migratory index in a realistic environment.

R2- Comment #8: Page 27- it has been mentioned that “the stiffness of cancer cells was assumed (277.3 Pa) based on previously reported by Corbin et al. [59] by AFM measurements”. Did authors note that AFM measures local stiffness (surface of the cell) not the bulk stiffness? While the provided model is based on the bulk stiffness why authors considered local stiffness measured by AFM instead of using bulk stiffness of cells which has been reported in several papers.

The AFM could provide both bulk and local stiffness values depending on the type of AFM indenters. Sharp indenters are mostly used to locally deform the membrane while in the mentioned study, they used a spherical indenter with size of 4.5 μm which could deform the whole cell and provide the bulk stiffness. In our calculations, we used the average values measured by AFM (277.3 Pa). Moreover, in that paper, the authors used a MEMS-based sensor by considering cells as a point mass with constant stiffness and viscosity to measure the rheological properties of cells. They assumed that cells have a bulk stiffness and their result are in good agreement with AFM results. For example for the same cell line (277.3 Pa), they reported 257 Pa elasticity. Their MEMS-based results even are in better agreement with our model as for highly invasive cancer cells, the bulk stiffness value is closer to the optimal range. For all three cell lines, the trend is the same and there are no big differences between results.

Action take:

According to the remark, in the relevant parts, instead of AFM values, we used MEMS-based values for discussion because these data are closer to the bulk stiffness of cells. Please see changes below:

Page 25: “Error! Reference source not found.c) which is in good agreement with the average bulk elasticity of the highly invasive human breast cancer cells (257.5 Pa) previously reported by Corbin et al. [60] using AFM measurements a MEMS resonant sensor. Based on the same

study, the average elasticity of the non-invasive human breast cancer cells (MCF-10A) is ~398 Pa, which is far from the critical range while reaches down to 257.5 Pa for MDA-MB-231[60]. The same study showed that the average elasticity of invasive cells (MCF7) is somewhere between two other cells line, but very close to the highly invasive cell lines (275.2 Pa)”

R2- Comment #9: Authors examined the effect of substrate stiffness, but such study can be useful when the substrate is considered as a deformable material. The migration behavior of cells strongly depends on substrate stiffness and its deformation. How authors address this point?

We studied the effects of substrate stiffness, and once the elasticity is low, the substrate is more deformable. The deformation of the substrate is not independent of its stiffness. In our simulation, the deformation of both cell and substrate were considered to determine the migratory index, and we discussed the results in the relevant section (1.3.3). We assumed cells are connected to a substrate (a rectangle) with different elasticities, ranging from 100 to 10000 kPa. We are uncertain what the reviewer meant with “deformable material”. We considered the substrate as a deformable material and to measure the migratory index, we calculated the displacement (deformation) at the cell-substrate interface. Initially, the cells were assumed to be on an un-deformed substrate, and once the contractile force is generated both cell and substrate parts deform depending on their elasticity. We did not measure the migratory index on substrates with different shapes to make a fair comparison on the stiffness of cells.

R2- Minor Comments:

- Page 8, defining the rigidity index requires a reference.

We defined the rigidity index by equation 1 to compare the elasticity of cancer cells with their normal counterparts. Negative values show softer cancer cells and positive values show the stiffer cancer cells.

Page 9: “For cancer or ~~invasive~~-malignant cells, the rigidity index could be defined with respect to the elasticity of their normal or ~~non-invasive-malignant~~ counterpart cells as follow:”

- The section numbering need to be modified starting from 1 for introduction, then 2. Method (including 2.1, 2.2, etc.)

Thanks for the comment. We edited the numbering and merged those parts to the method section.

- There is no need to define cytoplasm in page 11

As per your suggestion, we removed that sentence for the definition of cytoplasm.

- I suggest to add one table summarize all input parameters

We added a table (Table.1) to summarize all parameters used in our simulation and the other parameters can be found in the uploaded data.

- Some figures can be merged such as figure 6 and 7 together or figure 8 and 9 together.

We merged Figure 6 and 7 to “**Figure 6**”, and we merged Figure 8 and 9 to “**Figure 7**”

Rev 0	Fig. 1	Fig. 2	Fig. 3	Fig. 4	Fig. 5	Fig. 6	Fig. 7	Fig. 8	Fig. 9	Fig. 10
Rev 1	Fig. 1	Fig. 2	Fig. 3	Fig. 4	Fig. 5	Fig. 6		Fig. 7		Fig. 8

- Section 1.6 Migratory Index Model for Developing Anti-Cancer Drugs, is a part of discussion and does not have separate results, so there is no need to have a separate numbering for that.

We added a separated section, “Discussion”, and included the mentioned section as a sub-section of the discussion.

Reviewer 3 (R3)

Comments to the Author(s)

In this manuscript, these authors defined a quite interesting energy-depend term, a migratory index which could indicate cell migration potential, to explain the contradiction about whether the speed of cell migration and invasion were inversely correlated to cell stiffness. Their mechanical model showed that cells had a maximum migratory index taking place at a specific range of cell bulk elasticity by control cell physical parameters, indicating the maximum capability of cell migration. The authors also offered valuable advice about adopting different cell stiffness adjustment strategies to control the migration of cancer cells. Nevertheless, they needed more information to explain some terms and the Migration Index formula. More specific comments are detailed below:

R3-Comments # 1: The authors needed more explanation in “non-invasive counterparts” in the section of abstract “We believe that the stiffness of invasive cells depending on the stiffness of their non-invasive counterparts is either decreased or increased to reach the critical condition in which the mobility potential of cells is approximated to be maximum.” and introduction “Where r is the rigidity index, E_c is the elasticity of cancer or invasive cells, and E_n is the elasticity of the normal or non-invasive counterparts with the same unit.” What did non-invasive counterparts mean? Was it the substrate or surrounding cells? Or was it normal cells before cell became more invasive? If non-invasive counterparts meant normal cells, how to explain the invasive and non-invasive cells in human kidney, bladder, and thyroid cell in fig 1?

Thanks for pointing out this concern. We believe that the migratory index model could be used to predict the migratory behavior of cells in different stages, and that is why we used “invasive” and “non-invasive” cells. Based on the literature, invasive cells are migratory and non-invasive cells are less migratory. “Non- invasive” could either refer to a normal cell before becoming cancerous or benign or non-malignant cells before becoming malignant cells. “Invasive” could either refer to the cancer cells (when changing from normal to cancer) or malignant cells (when changing from benign cells to malignant cells). Based on the available data in the literature, cancer or malignant cells are more invasive than normal or non-malignant cells.

Action taken:

To avoid any confusion, we revised the mentioned sections by changing “non-invasive” to “normal and benign cells” and “invasive” to “malignant”.

Page 2:

“We believe that the stiffness of cancer or malignant ~~invasive~~ cells depending on the stiffness of their normal or non-malignant ~~non-invasive~~ counterparts is either decreased or increased to

reach the critical condition in which the mobility potential of cells is approximated to be maximum.”

Page 9:

“For cancer or ~~invasive~~-malignant cells, the rigidity index could be defined with respect to the elasticity of their normal or ~~non-invasive-malignant~~ counterpart cells as follow”

R3-Comments # 2: In the section of the introduction, “Beyond that range, cells might lose their capability to migrate fast and potentially move into other parts of the body.” If cells lost their capability to migrate fast, how could they move into other parts of the body?

We meant that cells lose their potential to migrate fast and invade other parts of the body.

Action taken:

To avoid any confusion, we revised that sentence.

Page 8: “Beyond that range, cells might lose their capability to migrate fast and have less migratory potential to move into other parts of the body”

R3-Comments # 3: In the section of “1.21 Cell Modeling and Force Generation”, Was it contradictory to assume that the bulk stiffness was constant and independent of contraction force and that cells were considered as a pure elastic mechanical body with a bulk stiffness?

In our modeling, we assumed that cells are pure elastic mechanical bodies to simplify our simulation. Biological parameters could affect the mechanical properties (stiffness) and consequently the migratory potential of cells, but here we tried to see the migration process from mechanics point of view. We assumed that the stiffness of cells doesn't change over migration and when the internal force is generated. So, the cells could be considered as a pure mechanical body. Then, a single cell is a pure elastic mechanical body with a constant bulk stiffness. Changes in the cytoskeletal organization during force generation might affect the stiffness, and we used this assumption to simplify our modeling. We already emphasized on this assumption on page 13 second paragraph.

Action taken:

We revised that sentence as below (page 11)

“..bulk stiffness of cells are independent. **Considering these assumptions, cells were modelled as a pure mechanical body with a bulk stiffness[28].**”

R3-Comments # 4: In the section of “1.32 Migratory Index (μ)”, the authors defined a formula for calculation the migratory index (μ), what did the adhesion complexes mean? At the same time, the authors needed more explain the form of this formula.

Adhesion complexes are connections (proteins) that integrate the cells to the extracellular matrix (substrate). Through these connections, internal forces can be transmitted to the cell-substrate interface and help cells move forward. Once the internal force is generated, depending on the shape and elasticity of them, energy could be stored into them. (Please see our response to **R2**)

Action taken:

As R2 suggested, we moved that section to the method parts and added the below details to explain the formula.

“Adhesion complexes are connections which integrate cells to their microenvironment (**Error! Reference source not found.**a). These thin structures have a critical role in migration and clinically are approved to drive cancer cell metastasis[50]. So, the migratory energy can be specifically computed at these adhesion complexes, migratory index, by integrating the dot product of stress vectors with strain vectors over the area of the connections to approximate the migratory capacity of cancer cells:

$$\mu = \frac{\iint \varepsilon \cdot \sigma dA_{adhesion\ complexes}}{2A_{adhesion\ complexes}} \quad 3$$

Where A is the total area of the connecting regions to the substrate, ε is strain field and is σ stress field in connecting regions and μ is the energy stored density. This term represents the energy density”

R3-Comments # 5: In the section of “1.33 Effects of substrate elasticity on the migratory index” and Fig 9, the authors needed a better explanation that the capability of cells for migration is boosted by increasing the substrate rigidity. Was it related to the value of μ or the peak position? If cell migration related to the value of μ , was the migration capacity of MDA-MB-231 cells on a 10 Pa substrate less than the migration capacity of MCF-10A on 100 kPa?

The migratory of cells can be increased by the value of the migratory index. More migratory index, more potential to migrate fast. On each substrate (with a specific elasticity), there may be a different optimal range. Generally, under the same condition, cells could migrate faster on the stiffer microenvironment. Each cell line might have different microenvironments. Our model shows that the “optimal range” could be influenced by the elasticity of the substrate, but it does not compromise the fact of increasing migration on the stiffer substrate. We compared the migratory potential and migratory index of cells under a same condition (Figure 7), and discussed different parameters affecting the migratory index. For this specific question, the migratory index should be calculated for both conditions. But based on the calculation discussed in the manuscript, the model predicts a higher capacity of migration for MDA-MB-231 compared to MCF-10A on a substrate with a same stiffness value. In fact, both cell stiffness and substrate stiffness could influence the migratory index.

Action taken:

We made small changes in the relevant section. On page 24, we clearly mentioned that the migratory index values (Figure 7c) are increased and allow cells to migrate faster. Then, we explained that there is a different optimal range depending on the substrate stiffness.

“It clearly shows that the migratory index **values** are increased by increasing the substrate stiffness...”

“...So, a stiffer substrate could enhance the mobility potential of cells (**migratory index**), and **allows them** to migrate faster...”

“..it can be observed that the migratory index peak is shifted to the right by stiffening the substrate, suggesting that **the critical stiffness range (peak) migration-of-cancer-cells** could be regulated by altering the mechanical properties of the extracellular matrix as well..”

“..In fact, for each substrate stiffness, there is a different critical **range of stiffness for cells..**”